The tetrapod fauna of the upper Permian Naobaogou Formation of China: 10. Jimusaria monanensis sp. nov. (Dicynodontia) shows a unique epipterygoid

Shi Yu-Tai 1 2
http://orcid.org/0000-0003-2205-9671 Liu Jun 1 2 liujun@ivpp.ac.cn
1 Key Laboratory of Vertebrate Evolution and Human Origins of Chinese Academy of Sciences, Institute of Vertebrate Paleontology and Paleoanthropology, Chinese Academy of Sciences , Beijing , China
2 College of Earth and Planetary Sciences, University of Chinese Academy of Sciences , Beijing , China
Marsicano Claudia
Electronic publication date: 2023 Jul 31
Publication date: 2023
Volume: 11
Electronic Location ID: e15783
Received 2023 May 19; Accepted 2023 Jul 3
Copyright: © 2023 Shi and Liu
Copyright year: 2023
Copyright holder: Shi and Liu
License: This is an open access article distributed under the terms of the Creative Commons Attribution License, which permits unrestricted use, distribution, reproduction and adaptation in any medium and for any purpose provided that it is properly attributed. For attribution, the original author(s), title, publication source (PeerJ) and either DOI or URL of the article must be cited.
License URL: https://creativecommons.org/licenses/by/4.0/

Keywords: Dicynodontia, Dicynodontoidea, Jimusaria, Lopingian, Naobaogou formation, Epipterygoid

Funding: Chinese Academy of Sciences XDB26000000 This work is supported by Strategic Priority Research Program of Chinese Academy of Sciences (XDB26000000). The funders had no role in study design, data collection and analysis, decision to publish, or preparation of the manuscript.

==============================
Jimusaria is the first reported Chinese dicynodont, previously only known from Xinjiang. Here we refer two specimens from the Naobaogou Formation, Nei Mongol, China to Jimusaria based on the following features: squamosal separated from supraoccipital by tabular, tabular contacting opisthotic, sharp and thin lateral dentary shelf expanding anteriorly into a thick swelling, nasals fused as single element, rod-like medial bar formed by footplate of epipterygoid connecting to the parabasisphenoid and periotic medially. A new species, J. monanensis, is named based on the diagnostic characters on these two specimens such as distinct caniniform buttress lacking posteroventral furrow, naso-frontal suture forming an anterior directed sharp angle, and converging ventral ridges on posterior portion of anterior pterygoid rami. In Jimusaria, the epipterygoid posteromedially contacts the parabasisphenoid and the periotic as a rod-like bar, a unique morphology unknown in any other dicynodonts. This structure probably increases the stability of the palatal complex. A similar structure might also appear in other dicynodonts as a cartilage connection. The new occurrence of Jimusaria increases the diversity of the tetrapod assemblage from the Naobaogou Formation, and further strengthens the connection between the tetrapod faunas from Nei Mongol and Xinjiang. Based on the current record, Jimusaria is one of the few tetrapod genera which survived in the end-Permian mass extinction.

Introduction

During the Sino-Swedish expedition, Yuan & Young (1934) collected some dicynodont fossils from Xinjiang, China. Dicynodon sinkianensis is the first named species from those specimens, and is the first record of Chinese Permian-Triassic tetrapods. Later, this species was transferred to a new genus, Jimusaria, resulting in a new combination, J. sinkianensis (Sun, 1963). Three incomplete skulls from Turpan, Xinjiang were also referred to this genus as J. taoshuyuanensis (Sun, 1973). King (1988) proposed Jimusaria as a junior synonym of Dicynodon and resumed the name D. sinkianensis, creating a new combination D. taoshuyuanensis. Lucas (1998, 2001) and Li, Wu & Zhang (2008) followed this opinion. Kammerer, Angielczyk & Fröbisch (2011) completed a comprehensive phylogenetic analysis on Dicynodon-related species; they revalidated Jimusaria, and regarded J. taoshuyuanensis as a junior synonym of J. sinkianensis.

Abundant tetrapod fossils have been collected from the Naobaogou Formation of Nei Mongol, and they are referred to diverse clades such as Chroniosuchia, Captorhinidae, Pareiasauria, Dicynodontia, and Therocephalia (Li & Cheng, 1997; Liu, 2021, 2023; Liu & Abdala, 2020; Liu & Bever Gabriel, 2018; Liu & Chen, 2021; Zhu, 1989). This assemblage is dominated by the dicynodonts, including Daqingshanodon limbus and Turfanodon jiufengensis. Turfanodon is the first described dicynodont genus shared by Nei Mongol and Xinjiang and distributed across both tropical and temperate zones (Liu, 2021). As mentioned but not described in Liu (2019), other than Turfanodon, Jimusaria was also produced from the Naobaogou Formation. Here we describe two specimens and establish a new species of Jimusaria, this new species further increases the diversity of the tetrapod assemblage of the Naobaogou Formation and the connection between the North China and Xinjiang during the Lopingian, Permian.

Materials and Methods

IVPP V26034, a nearly complete skull with mandible, six cervical vertebrae and an incomplete right scapula articularted with an incomplete bone (? coracoid); the skull basal length is 223 mm (Figs. 1 and 2). IVPP V31929, an incomplete skull with complete mandible; the skull basal length is 143 mm (Figs. 3, 4A, 4E and 5).

Figure 1 Jimusaria monanensis sp. nov. from the Naobaogou Formation, holotype, IVPP V26034.

(A) Dorsal; (B) ventral views. AN, angular; AR, articular; bt, basal tuber; BO, basioccipital; co, crista oesophagea; D, dentary; F, frontal; ipv, interpterygoid vacuity; J, jugal; L, lacrimal; la, lacrimal fossa; M, maxilla; N, nasal; P, parietal; PBS, parabasisphenoid; PE, periotic; PF, prefrontal; pif, pineal foramen; PL, palatine; PM, premaxilla; PO, postorbital; PP, preparietal; PRE, prearticular; PT, pterygoid; ptf, post-temporal fenestra; Q, quadrate; SP, splenial; SQ, squamosal; t, tusk; V, vomer. Scale bar represents 5 cm. Photo credit: Gao Wei.

Figure 2 Jimusaria monanensis sp. nov. from the Naobaogou Formation, holotype, IVPP V26034.

(A) Right lateral; (B) posterior views; (C) photo of vertebrae and scapula. AN, angular; AR, articular; COR, coracoid; D, dentary; EPI, epipterygoid; F, frontal; fm, foramen magnum; J, jugal; L, lacrimal; la, lacrimal foramen; M, maxilla; mf, mandibular fenestra; N, nasal; ntn, notch for the trigeminal nerve; oc, occipital condyle; P, parietal; PBS, parabasisphenoid; PE, periotic; PF, prefrontal; PL, palatine; pla, pila antotica; PM, premaxilla; PT, pterygoid; ptf, post-temporal fenestra; PO, postorbital; pop, paroccipital process; Q, quadrate; QJ, quadratojugal; SA, surangular; SC, scapula; SM, septomaxilla; SO, supraoccipital; SQ, squamosal; t, tusk; V, vomer. Scale bars represent 5 cm. Photo credit: Gao Wei.

Figure 3 Jimusaria monanensis sp. nov. from the Naobaogou Formation, IVPP V31929.

(A) Ventral; (B) anterior views. AN, angular; AR, articular; BO, basioccipital; co, crista oesophagea; D, dentary; EC, ectopterygoid; EPI, epipterygoid; J, jugal; ic, internal carotid canal; ipv, interpterygoid vacuity; M, maxilla; PBS, parabasisphenoid; PE, periotic; PL, palatine; PM, premaxilla; PT, pterygoid; pr, posterior ridge of premaxilla; PRE, prearticular; Q, quadrate; ST, stapes; SP, splenial; SQ, squamosal; t, tusk; V, vomer. Scale bar represents 5 cm. Photo credit: Gao Wei.

Figure 4 Jimusaria monanensis sp. nov. from the Naobaogou Formation, (A) and (E) IVPP V31929.

J. sinkianensis from the Guodikeng Formation, (B) V3240.1; (C) and (F) IVPP RV341407; (D) V3240.2. (A)--(D) in dorsal views; (E) and (F) in occipital views. BO, basioccipital; EPI, epipterygoid; EO, exoccipital; F, frontal; jf, jugular foramen; N, nasal; P, parietal; PE, periotic; PF, prefrontal; POF, postfrontal; pop, paroccipital process; PPA, postparietal; ptf, post-temporal fenestra; SO, supraoccipital; SQ, squamosal; TA, tabular. Scale bars represent 5 cm. (A), (C) (E) and (F) photo credit: Gao Wei. (B) and (D) photo credit: Yu-Tai Shi.

Figure 5 Jimusaria monanensis sp. nov. from the Naobaogou Formation, IVPP V31929.

(A) Right lateral; (B) left lateral views; (C) drawing of mandible in left lateral view. AN, angular; AR, articular; D, dentary; EC, ectopterygoid; F, frontal; J, jugal; L, lacrimal; lds, lateral dentary shelf; M, maxilla; mf, mandibular fenestra; N, nasal; OS, orbitosphenoid; PBS, parabasisphenoid; PE, periotic; PF, prefrontal; pif, pineal foramen; PL, palatine; PM, premaxilla; PRS, presphenoid; PT, pterygoid; Q, quadrate; QJ, quadratojugal; SA, surangular; SM, septomaxilla; smf, septomaxillary foramen; SQ, squamosal; t, tusk; V, vomer. (B) and (C) share same scale bar. All scale bars represent 5 cm.

Phylogenetic analysis

The matrix is based on the recent works (Angielczyk, Liu & Yang, 2021; Kammerer & Ordoñez, 2021; Liu, 2022). The final data set consists of 199 characters (23 continuous and 176 discrete-states) and 120 species (OTUs). Continuous characters were treated as additive, and eight discrete-state characters were treated as ordered (characters 56, 81, 84, 102, 163, 173, 189 and 199). The data were analyzed using parsimony in TNT v1.6 (Goloboff & Morales, 2023) using New Technology search parameters, starting at level 65 and forced to find the shortest tree at least 50 times checking every five hits. Then using the traditional search method analysis trees produced from the New Technology search. Biarmosuchus served as the outgroup to root the most parsimonious tree in our analyses. Symmetric resampling values were calculated based on 10,000 replicates.

Nomenclatural acts

The electronic version of this article in Portable Document Format (PDF) will represent a published work according to the International Commission on Zoological Nomenclature (ICZN), and hence the new names contained in the electronic version are effectively published under that Code from the electronic edition alone. This published work and the nomenclatural acts it contains have been registered in ZooBank, the online registration system for the ICZN. The ZooBank LSIDs (Life Science Identifiers) can be resolved and the associated information viewed through any standard web browser by appending the LSID to the prefix http://zoobank.org/. The LSID for this publication is: urn:lsid:zoobank.org:pub:B5D7A3C5-1C07-4F71-BC24-9141CD5B658E. The online version of this work is archived and available from the following digital repositories: PeerJ, PubMed Central SCIE and CLOCKSS.

Systematic palaeontology

Anomodontia Owen, 1860

Dicynodontia Owen, 1859

Dicynodontoidea Olson, 1944

Genus Jimusaria Sun, 1963

Type species. Jimusaria sinkianensis (Yuan & Young, 1934).

Revised diagnosis. A medium-sized dicynodontoid with narrow intertemporal bar, zygomatic and quadrate rami of squamosal forming an acute angle. It is also differentiated from other dicynodontoids by squamosal separated from supraoccipital by tabular contacting opisthotic; sharp and thin lateral dentary shelf expanding anteriorly into a thick swelling; distinct medium-sized caniniform process; nasals fused as single element; rod-like medial bar formed by footplate of epipterygoid, connecting to the parabasisphenoid and periotic medially.

Jimusaria monanensis sp. nov.

Etymology. ‘Monan,’ means “south of the Gobi Desert of Mongolia”.

Holotype. IVPP V26034, a nearly complete skull with mandible, six cervical vertebrae, an incomplete right scapula articulated with an incomplete bone (? coracoid) (Figs. 1 and 2).

Type locality and horizon. Tumed Right Banner, Nei Mongol, China; Member I, Naobaogou Formation.

Referred material. IVPP V31929 Shiguai, Baotou, Nei Mongol, China. Member I, Naobaogou Formation, an incomplete skull with complete mandible (Figs. 3, 4A, 4E and 5).

Diagnosis. Differentiated from type species by distinct caniniform buttress lacking posteroventral furrow, naso-frontal suture forming an anterior directed sharp angle, and converging ventral ridges on posterior portion of anterior pterygoid rami.

Description

Skull

IVPP V26034 and V31929 are different in size. IVPP V26034 may represent an adult. Its skull is slightly crushed and the both postorbital bars are broken; while the left ramus of the mandible is broken. In IVPP V31929, the incomplete skull has a weathered skull roof, and the right zygomatic arc and left squamosal are missing (Figs. 3, 4A, 4E and 5). The following description is based on the both specimens if not specified.

The short snout has a smoothly curved dorsal surface, and its anterior surface formed by the premaxilla is low and relatively flat. Its ventral margin has a shallow middle invagination (Fig. 3B). The snout region around the external nares is well-sculptured in IVPP V26034, but is only slight rugose in IVPP V31929 (Figs. 1A, 2A, 3B and 5).

The single premaxilla forms most part of the secondary palate. On the palatal surface, there are a pair of parallel anterior ridges and a posterior ridge. The anterior ridges extend to the anterior premaxillary margin. The narrow posterior ridge extends anteriorly to the level of the tusk, and posteriorly to the vomer, which has nearly the same width (Figs. 1B and 3A).

The septomaxilla lies within the anterodorsal concavity of the maxilla and forms the posterior margin of the naris. It contacts the premaxilla anteriorly and the nasal dorsally (Figs. 2A, 5A and 5B). In IVPP V31929, a septomaxillary foramen lies on the right side of the septomaxilla-maxilla junction (Fig. 5A). The postnarial region does not bear an excavation.

In lateral view, the maxilla contacts the septomaxilla and the premaxilla anteriorly, the nasal and the lacrimal dorsally, the jugal posterodorsally, and the squamosal posteriorly (Figs. 2A, 5A and 5B). It forms a regular-sized caniniform process as in most Permian dicynodonts. The caniniform process expands laterally as a distinct buttress, which bears a flat posterior surface without the lateral furrow as in J. sinkianensis (Kammerer, Angielczyk & Fröbisch, 2011). The ventral edge of the caniniform process lies at the level of the posterior margin of the naris. The round tusk is curved posteriorly, and the base is directed anteroventrally while the tip is directed posteroventrally.

The nasals are well-fused without a median suture, same as in J. sinkianensis (Yuan & Young, 1934) (Figs. 1A and 4A). The nasal boss is developed as a median swelling, forming the posterodorsal margin of the naris. The nasal contacts the prefrontal posterolaterally and the frontal posteriorly, the naso-frontal suture has two straight segments which form a sharp angle (Figs. 1A, 4A and 4C).

The round orbit faces more laterally than anteriorly (Fig. 3B). The orbit is formed by the flat prefrontal anterodorsally, the lacrimal anteriorly, the jugal ventrally, the postorbital posteriorly, the frontal dorsally, and the postfrontal posterodorsally (Figs. 1A, 2A and 5A). The lacrimal foramen lies close to the suture with the prefrontal. The jugal extends posteriorly behind the postorbital bar and contributes to the zygomatic arch, but it has a little contribution to the postorbital bar (Figs. 1B and 3A).

The frontals have distinct anteromedial processes (Figs. 1A and 4A). The left postfrontal can be identified in IVPP V31929, its surface is flat, without the groove as in IVPP RV341407 (Figs. 4A and 4C). The oval preparietal is flush with the skull roof, and forms the anterior margin of the pineal foramen (Figs. 1A and 4A).

Of the preserved postorbital bars, the only complete one is on the left side of IVPP V31929. The postorbital bar is almost entirely formed by the postorbital (Figs. 2A, 5A and 5B). The ventral portion of the postorbital bar is extremely anteroposteriorly expanded and mediolaterally flattened. The postorbital ascends dorsally and medially to join the skull roof, then extends posteriorly and forms the entire intertemporal bar with the parietal. In addition to the postorbital, the slender parietal also contributes to the narrow intertemporal bar, and it has a narrow dorsal exposure within the midline groove. The poor preservation obscures whether a distinct sagittal crest was present (Fig. 1A). The postparietal contributes to the intertemporal bar but is covered by the postorbital dorsally.

The right squamosal of IVPP V26034 is relatively complete unlike the other (Fig. 2A). The squamosal has a long zygomatic ramus, which has a flattened posterior portion, a long sutured area with the jugal, and bears a pointed anterior tip inserting to the maxilla below the orbit. The lateral fossa formed by the zygomatic ramus and quadrate ramus of the squamosal can be observed in posterior view. The quadrate ramus of the squamosal descends ventrally with lateral expansion. The dorsal portion of the quadrate ramus is curved posteriorly. The quadrate ramus receives the fused quadratojugal and quadrate on its anteroventral surface (Figs. 2A and 5A). The quadrate foramen, between the quadrate and quadratojugal, faces anteromedially. The quadrate, articulating to the articular of the mandible, has a typical W-shape facet.

The narrow vomer forms the septum and posterior margin of the choana (Figs. 1B and 3A). The interpterygoid vacuity is formed by the vomer and the pterygoid. Its posterior margin is flush with the median pterygoid plate. The vomer raises dorsally above the choana as a vertical sheet which laterally clasps the parasphenoid (Figs. 5A and 5B).

The palatine widens anteriorly to form a rugose palatine pad, which contacts the premaxilla and the maxilla. Posteriorly, the palatine forms the lateral wall of the choana with the pterygoid. Laterally, the labial fossa is surrounded by the jugal laterally, the palatine medially, and the maxilla anteroventrally. The ectopterygoid also contributes to the labial fossa margin. The ectopterygoid contacts the maxilla anteriorly and the pterygoid posteriorly, extending posteriorly without exceeding the posterior margin of the palatine pad in ventral view (Fig. 3A).

The pterygoid shows the typical X-shape found within the dicynodonts (Figs. 1B and 3A). The anterior ramus of the pterygoid touches the maxilla anteriorly. The anterior pterygoid ridges extend for most of the length of the anterior ramus of the pterygoid; the posterior portion of that ridge converges posterior to the interpterygoid vacuity, differing from J. sinkianensis (Yuan & Young, 1934) (Fig. 6). The median pterygoid plate has a distinct and thin crista oesophagea, which extends anteriorly to the posterior margin of the interpterygoid vacuity and bifurcates into two low rami posteriorly.

Figure 6 Photos of all specimens of Jimusaria sinkianensis.

(A) IVPP RV341407; (B) V3240.1; (C) V3240.2; (D) V3240.3 in ventral views; the arrows indicate the autapomorphies of J. sinkianensis which are the caniniform process with posterior furrow and the parallel ventral ridge on posterior portion of anterior pterygoid rami. (B–D) share the same scale bar. Scale bars represent 5 cm.

The parabasisphenoid bears paired internal carotid canals, whose ventral openings are located near the suture with the pterygoid (Fig. 3A). The parabasisphenoid contributes to the anterodorsal margin of the basal tuber as a ridge in a very steep angle. The posterior portion of the basal tuber is formed by the basioccipital (Figs. 1B, 3A and 4E). The basal tuber is elongated anteroposteriorly and laterally directed with relatively narrow edges. The intertuberal ridge connects the paired basal tubera (Figs. 1B and 3A). The parabasisphenoid is well exposed in lateral view in IVPP V31929 (Figs. 5A and 5B). It contacts the periotic posteriorly, and extends anterodorsally. Its anterior portion supports the presphenoid dorsally. The parabasisphenoid and the presphenoid are fused as a plate here. The orbitosphenoid is partly preserved, it separates into two wings which contact the frontal dorsally.

The incomplete left stapes is preserved between the quadrate and the basal tuber in IVPP V31929 (Fig. 3A). The dorsal process is present. It has an “extra facet”, similar to Kingoria (Cox, 1959) and Daptocephalus (Ewer, 1961).

The epipterygoids are not so well preserved, a dorsal end is observed on the right side of IVPP V26034 (Fig. 2A) and a left footplate rests on the quadrate ramus in IVPP V31929. Posteriorly, the footplate extends medially forming a rod-like medial bar, contacting the parabasisphenoid and the periotic suture (Figs. 3A and 7). Examining the holotype of J. sinkianensis, the epipterygoid also contacts the lateral braincase in a parabasisphenoid-periotic suture (Fig. 7B). This character is not reported in any other dicynodonts.

Figure 7 Photos of Jimusaria monanensis and J. sinkianensis.

(A) IVPP V31929 and (B) RV341407 in anterolateral views; (C) the braincase of V31929 in lateral view; (D) the drawing of the braincase in lateral view; the arrows indicate the connection between the epipterygoid and the lateral braincase wall. EPI, epipterygoid; PBS, parabasisphenoid; PE, periotic; PRS, presphenoid; PT, pterygoid; SQ, squamosal. Scale bars represent 2 cm.

In IVPP V31929, the periotic is composed of the prootic and the opisthotic (Figs. 4E and 5A), while the occipital elements are separated. In the holotype, it is unknow if the occipital elements are fused to the periotic due to the poor preservation (Fig. 2B). In IVPP V26034, the rod-like pila antotica rises anterodorsally from the periotic and forms a deep V-shaped notch for the trigeminal nerve (Fig. 2A); the notch for the vena capitis dorsalis is partly preserved on the periotic anterior edge. Laterally, the supraoccipital separates the periotic from the parietal (Fig. 2A).

No bone suture can be identified on the occipital plate of the holotype, but some sutures are traceable on IVPP V31929 (Fig. 4E). The postparietal is exposed on the occipital plate. It is supported by the supraoccipital ventrally, and contacts the tabular laterally. The supraoccipital is a large element forming the dorsal half margin of the foramen magnum. The tabular separates the supraoccipital from the squamosal, it extends ventrolaterally to the margin of the post-temporal fenestra, contacting the periotic ventromedially. The post-temporal fenestra is almost entirely formed by the periotic medially and the squamosal laterally. The paired exoccipitals meet on the midline and forms the ventral half of the foramen magnum, they join the basioccipital to form the tri-radiate occipital condyle. The facet for the proatlas is developed on the dorsal margin of the exoccipital, the jugular foramen lies near the occipital condyle surrounded by the exoccipital.

Mandible

The mandible is articulated with the skull, and there is one relatively complete ramus in each specimen (Figs. 2A, 5B and 5C). The dentary symphysis forms an upturned beak, with a smooth anterior surface (Figs. 2A, 3B and 5C). The anterior surface has a clear border with the lateral surface. The dorsal surface of the dentary symphysis comprises a medial groove and two lateral tables, on which the medial edge is higher than the lateral edge. The posterior dentary sulcus is narrow. Laterally, a distinct but thin lateral dentary shelf is situated on the dorsal edge of the mandibular fenestra, extending for almost the entire length of the fenestra and expanding into a prominent swelling anterodorsally (Fig. 5C). The dentary posteroventral process extends on the lateral surface of the angular, ventral to the mandibular fenestra. The angular joins the symphysis with its anterior tip. It also contributes to the ventral margin of the mandibular fenestra (Figs. 1B, 2A, 3A and 5C). All the reflected laminae are incomplete. The angular gap is wide; a ridge runs dorsoposteriorly from the dorsal notch (Figs. 2A and 5C), as Daqingshanodon limbus, Taoheodon baizhijuni and Kunpania scopulusa (Angielczyk, Liu & Yang, 2021); and the posteroventral fossa can be observed in IVPP V31929 (Figs. 5B and 5C). Those reflected lamina’s anatomical terminologies were followed by Olroyd & Sidor (2022). The surangular lies dorsal to the angular and contacts the dentary anteriorly (Figs. 2A and 5C). The articular provides an articular facet with the quadrate, which allows the parasagittal movement.

The splenial forms the posteroventral portion of the symphysis. It can be observed in medial or ventral view. The prearticular extends anteriorly from the articular, forming the ventral portion of the mandible ramus on the medial surface, it contacts the surangular dorsally, the dentary anterodorsally, and splenial anteriorly (Figs. 1B and 3A).

Postcranial bones

The holotype preserves seven articulated cervical vertebrae, beginning with the atlas, an incomplete right scapula and possiblely part of the coracoid (Fig. 2C). Because the cervical number is no more than seven in anomodonts (Fröbisch & Reisz, 2011; Liu, 2021), these vertebrae likely include all of the cervicals.

The atlas consists of two separate neural arches, the transverse process has a high dorsoventral expansion. The axis has an anteroposteriorly expanding neural spine, the articular facet of its postzygapophysis faces more laterally than ventrally. The odontoid is partly exposed in dorsal view, fused with the axis centrum. The 3rd vertebra has a nearly vertical neural spine, which narrows in anteroposterior width dorsally. Its postzygapophysis has a more or less horizontal articular facet. In the 5–6th vertebrae, the neural spines expand in width and incline slightly posteriorly. All centra are amphicoelous.

The bowed scapula is incomplete, with broken anterior and dorsal margins. Posteriorly, a rugose portion can be observed on the base of scapula, which represents the origination of the scapular head of M. triceps.

Discussion

In a previous study, IVPP V26034 was proposed to be closely related to Jimusaria sinkianensis (Liu, 2019). IVPP V26034 and V31929 can be referred to Jimusaria by the distinct medium-sized caniniform process, the sharp and thin lateral dentary shelf expanding anteriorly into a thick swelling, and fused nasals. However, these specimens show the following features which differentiated them from J. sinkianensis: anteriorly directed naso-frontal suture (Figs. 1A, 4A and 4B), the caniniform process without posterolateral furrow, and the keel of anterior rami of pterygoid converging posteriorly (Figs. 3A and 6). Hence, J. monanensis is established for them based on these diagnostic characters.

Jimusaria taoshuyuanensis was considered as a junior synonym of J. sinkianensis by Kammerer, Angielczyk & Fröbisch (2011). The referred specimens (IVPP V3240) bear the diagnostic features of J. sinkianensis, but are clearly different from the new species, supporting their conclusion. All specimens have a distinct caniniform process with a posterior furrow (Fig. 6), V3240.1 and V3240.2 expose a relative straight naso-frontal suture (Figs. 4C and 4D), V3240.2 and V3240.3 show paired parallel ventral ridges on the posterior portion of the anterior pterygoid rami (Fig. 6).

The holotype of Jimusaria sinkianensis shows that the tabular contacts the periotic, and separates the squamosal from the supraoccipital; the footplate of the epipterygoid elongates medially and contacts both the parabasisphenoid and the periotic as in J. monanensis (Figs. 4F and 7), so those characters are autapomorphies of Jimusaria. In J. sinkianensis, the parietal is narrowly exposed within the shallow groove of the intertemporal bar, and whether the postparietal contributes to the intertemporal bar is unknown in holotype (Fig. 4C), so the diagnosis of this species should be revised from Kammerer, Angielczyk & Fröbisch, 2011.

Phylogenetic analysis

The phylogenetic position of Jimusaria is highly unstable within Dicynodontoidea in previous studies. It was recovered outside the clade of Lystrosauridae plus Kannemeyeriiformes (Angielczyk, Hancox & Nabavizadeh, 2017; Angielczyk, Liu & Yang, 2021; Cox & Angielczyk, 2015; Kammerer, Angielczyk & Fröbisch, 2011; Kammerer, Fröbisch & Angielczyk, 2013; Kammerer & Smith, 2017), as the sister group of Gordonia and forming a clade with Sintocephalus plus Euptychognathus (Angielczyk & Kammerer, 2017; Kammerer, 2018), or being part of the monophyletic group (Jimusaria + Gordonia) as the sister group of Kannemeyeriiformes (Kammerer, 2019a; Kammerer, 2019b; Kammerer et al., 2019; Liu, 2021). It also was recovered more closely related to Dicynodon than Lystrosaurus (Olivier et al., 2019). Recent analyses recovered it in a relatively basal position within Dicynodontoidea (Kammerer & Ordoñez, 2021; Liu, 2022).

Jimusaria monanensis was coded based on IVPP V26034 and V31929 for the recent character list (Angielczyk, Liu & Yang, 2021; Kammerer & Ordoñez, 2021; Liu, 2022), some codings of J. sinkianensis were revised (Supplemental Files). Finally, one most parsimonious tree of length 1,273.225 was recovered (Fig. 8).

Figure 8 Phylogeny of Bidentalia.

Jimusaria monanensis in bold. Labeled clades: C, Cryptodontia; D, Dicynodontoidea. Numbers at nodes represent symmetric resampling values >45.

The topology is basically same as for Liu (2022) and Kammerer & Ordoñez (2021), but different from Angielczyk, Liu & Yang (2021) (Fig. 8). Kunpania was recovered as one of the basal members of Bidentalia. Jimusaria was recovered at the same position as for Liu (2022) and Kammerer & Ordoñez (2021), there is no monophyletic Jimusaria + Gordonia.

The epipterygoid morphology of Jimusaria

The posterior portion of the footplate sends the rod-like medial bar to the parabasisphenoid and periotic medially in IVPP V31919 (Figs. 7A and 7C). Ventrally, this rod seems like a connection between the quadrate ramus of the pterygoid and the lateral braincase wall in the pterygo-paroccipital foramen (Fig. 3A).

The footplate of the epipterygoid contacting the parabasisphenoid is a common character in dicynodonts, but that suture connection occurs between the anterior ramus of the footplate and the basipterygoid process, as observed in basal dicynodonts, such as Eodicynodon, Diictodon, and Pristerodon (Barry, 1968; Barry, 1974; Sullivan & Reisz, 2005), and bidentalians (Cluver, 1971; Keyser, 1973). This type of connection is quite different from that of Jimusaria.

A contact between the epipterygoid and the prootic is a relatively “derived” feature in therapsids. Some therocephalians possess a dumb-bell-shaped epipterygoid, and that dorsal expanding portion of the ascending ramus of the epipterygoid contacts the prootic (Barry, 1965; Durand, 1991; Huttenlocker, Sidor & Smith, 2011; Liu & Abdala, 2022). Some therocephalians and some cynodonts have a plate-like epipterygoid which sutures with the prootic posteriorly and forms part of the lateral wall of the braincase (Barry, 1965; Kemp, 1972; Olson, 1944). Some dinocephalians have an expanded epipterygoid which meets the periotic posteriorly too (Olson, 1944). However, none of those junctions are rod-like or originatie from the footplate of the epipterygoid (Barry, 1965; Olson, 1944).

Some therocephalians show a similar structure in a comparable portion of the skull, but the sheet-like structure just anterior to the post-temporal fenestra is formed by the otic process of the squamosal, the anteroventral process of the squamosal, or both. Although the epipterygoid also sends a ramus to that process (Durand, 1991; Kemp, 1972; Liu & Abdala, 2019; van den Heever, 1994). van den Heever (1994) deduced the function of the sheet-like process as protecting the neurovascular elements which pass through the post-temporal fenestra.

The location and shape of the rod-like medial bar of the epipterygoid in Jimusaria are quite different, and quite unlikely had same function. One possible function of the rod-like medial bar of the epipterygoid might be an additional auditory conduction medium, as some authors presumed that fossorial dicynodonts used the skull roof to sense quaking in the ground when they lived in underground burrows (Cox, 1972; Laaß, 2015). Under this situation the sound conducts through the skull roof to the inner ear, and the slender epipterygoid with the medial connection to the periotic could act as an efficient transmit route. The fossorial dicynodonts could show a complex naso-frontal suture and oblique stapes, however both the shape of the naso-frontal suture and the horizontal orientation of the stapes do not support it for Jimusaria (Kammerer, 2021; Laaß, 2015, 2016). Another possible function of the rod-like medial bar of the epipterygoid is increasing the stability of the palatal complex. In dicynodonts, the ventrolateral ridge of the parabasisphenoid is presumed to provide the attachment for M. protractor pterygoidei, which connects the parabasisphenoid and the quadrate ramus of the pterygoid, and keeps the palatal complex in its proper position (Surkov & Benton, 2004). In Jimusaria, the ventrolateral ridges of the parabasisphenoid are present, indicating M. protractor pterygoidei is developed, suggesting the demand of stabilizing the palatal complex in proper position. The epipterygoid in tetrapods generally has the function of stabilizing the palatal complex and connecting it to the skull roof (Romer, 1956), it is reasonable to assume the rod-like medial bar can synergize with M. protractor pterygoidei to enhance the stability of palatal complex.

The epipterygoid ossified from the palatoquadrate cartilage; representing an element of the splanchnocranium, in early tetrapods it connects to the quadrate frequently (Barry, 1965). In dicynodonts, the unossified palatoquadrate cartilage may be present dorsal to the pterygoid, from the plate of the pterygoid to the quadrate ramus of the pterygoid; as cartilage still links the epipterygoid and the quadrate in some living lizards (Jollie, 1960). This hypothesis can explain the variable position of the footplate of the epipterygoid in dicynodonts (Barry, 1974; Ewer, 1961; Maisch, 2002; Surkov & Benton, 2004). Therefore, there are two possible scenarios for the osteogenesis of the rod-like medial bar of the epipterygoid to the lateral wall of braincase: (1) the palatoquadrate has a unique medial projection and that portion ossifies as a part of the epipterygoid in Jimusaria; or (2) a cartilaginous connection is present between the quadrate ramus of the pterygoid and the lateral wall of braincase in most if not all dicynodonts, but the ossification occurred in Jimusaria. The second scenario seems more plausible considering the inferred function of the epipterygoid. The cartilaginous connection can provide a weaker but similar support function.

The ontogenetic variation of Jimusaria

IVPP V26034 differs from V31929 in size, and they are inferred to represent different ontogenetic stages. The smaller specimen (IVPP V31929) has a smoother snout region with only some pits and rugosity (Fig. 3B), while the larger specimen (IVPP V26034) has a distinctly rough snout (Figs. 1A and 2A). A similar pattern also occurred in Jimusaria sinkianensis (Figs. 4B–4D). The changing from pits to sculpture might indicate that the keratinous beak became heavier during the growth, as appeared in Turfanodon bogdaensis (Liu, 2021).

Comparing two specimens (IVPP V26034 and V31929), the interorbital region becomes relatively wider, and the intertemporal bar becomes narrower (although it might cause by deformation); the pterygoid medial plate is relatively longer while the interpterpterygoid vacuity is relatively smaller in the bigger specimen (IVPP V26034). The smaller specimen (IVPP V31929) already has a distinct caniniform buttress, indicating that feature developed at a relatively young age in Jimusaria monanensis.

In J. sinkianensis, three small specimens (IVPP V3240) show the distinct caniniform buttresses and the indistinct posteroventral furrow continues dorsoposteriorly to the depressions in the maxilla; the larger specimen (IVPP RV341407) possesses fatter caniniform processes with distinct posteroventral furrows and stronger depressions (Fig. 6). The tusks are directed vertically in the larger specimen rather than anteroventrally as in the small specimens. So, the growth pattern of the caniniform process is different between these two species.

Paleobiogeographic and biostratigraphic implications

The tetrapod assemblage of the Naobaogou Formation is the most diverse tetrapod assemblage of the Lopingian in China, the previous report of Turfanodon jiufengensis implies the Naobaogou Formation shares common taxon with the Guodikeng Formation (Liu, 2021). Several dicynodonts have been reported from the Guodikeng Formation of Xinjiang, including Jimusaria sinkianensis, T. bogdaensis, Diictodon feliceps, and some Lystrosaurus species (Li, Wu & Zhang, 2008). The finding of J. monanensis enhances the connection between the faunas of Nei Mongol and Xinjiang. Same as Turfanodon, Jimusaria is the second reported dicynodont genus which has a distribution across the temperate and tropic zones.

Recent stratigraphic work placed the Permo-Triassic boundary at the base of the Guodikeng Formation in Xinjiang (Yang et al., 2021). Based on the field notes, the horizon of the holotype of J. sinkianensis is the upper portion of the Guodikeng Formation, early Triassic in age (Angielczyk et al., 2022) while other specimens of this species also could be Triassic in age. Meanwhile, two specimens of J. monanensis were collected from Member I of the Naobaogou Formation, which possibly correlates with the Wuchiapingian (Shen et al., 2022, 2021). So, Jimusaria is a rare example of a tetrapod genus which survived in the end-Permian mass extinction.

Conclusions

A new dicynodont species, Jimusaria monanensis, was discovered in the Naobaogou Formation, this record further increases the diversity of the tetrapod assemblage of this formation and the connection between the North China and Xinjiang during the Lopingian, Permian. The peculiar rod-like medial bar of the epipterygoid in Jimusaria likely performed a function in stabilizing the palatal complex, and a similar structure may have been present in many other dicynodonts as part of palatoquadrate cartilage remnants. Jimusaria spanned from the Wuchiapingian to the Early Triassic in age.

Supplemental Information

Supplemental Information 1 The character list and notes on coding changes for phylogenetic analysis and matrix.

Click here for additional data file.

We gratefully thank the field crew of Daqingshan (Jia Zhen-Yan, Chang Shao-Ning, Wang Yu, Zhang Shao-Guang, and Liu Yu-Dong), especially Chang Shao-Ning and Liu Yu-Dong who discovered that two specimens. Specimens were prepared by Wu Yong and Fu Hua-Lin and the photos were taken by Gao Wei; Kenneth Angielczyk helped for interpreted the measure criterion of continuous character. We also sincerely thank Christian Kammerer, an anonymous reviewer and Leandro Gaetano, theirs comments and revisions of grammar improved this article.

INSTITUTIONAL ABBREVIATIONS

IVPP Institute of Vertebrate Paleontology and Paleoanthropology, Chinese Academy of Sciences, Beijing, China

Additional Information and Declarations

Competing Interests

Author Contributions

Data Availability

New Species Registration

The authors declare that they have no competing interests.

Yu-Tai Shi conceived and designed the experiments, performed the experiments, analyzed the data, prepared figures and/or tables, authored or reviewed drafts of the article, and approved the final draft.

Jun Liu conceived and designed the experiments, prepared figures and/or tables, and approved the final draft.

The following information was supplied regarding data availability:

The raw data are available in the Supplemental File.

All specimens are stored in IVPP (Institute of Vertebrate Paleontology and Paleoanthropology, Chinese Academy of Sciences, Beijing). Including IVPP V26034, a nearly complete skull with mandible, six cervical vertebrae and an incomplete right scapula articularted with an incomplete bone (? coracoid); and IVPP V31929, an incomplete skull with complete mandible.

The following information was supplied regarding the registration of a newly described species:

Publication LSID: urn:lsid:zoobank.org:pub:B5D7A3C5-1C07-4F71-BC24-9141CD5B658E.

Jimusaria monanensis sp. nov. LSID: urn:lsid:zoobank.org:act:87229D2E-90B1-4020-A7F9-C60F5785B55B.

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
