# Peer review of "The tetrapod fauna of the upper Permian Naobaogou Formation of China: 10. Jimusaria monanensis sp. nov. (Dicynodontia) shows a unique epipterygoid"

_PeerJ, doi:10.7717/peerj.15783_

## Round 0.1 · original submission · Minor Revisions

Dear Dr. Shi

You manuscript # 86077 entitled "The tetrapod fauna of the upper Permian Naobaogou Formation of China: 10. Jimusaria monanensis sp. nov. (Dicynodontia) with the unique epipterygoid", co-authored with Jun Liu, submitted to PeerJ has been reviewed by 3 reviewers and the editor.

All reviewers are coincident that your contribution is suitable for publication and should be accepted after minor revision. In this context, the reviewers have pointed out some changes concerning misspellings, minor changes in the figures, and rewording, among others. Two annotated versions of your manuscript with the suggested changes have been also included with the revisions.

Please, pay attention to suggestions made by all reviewers about the identification and careful description of the new taxon´s epipterygoid, which is crucial in your Ms discussion. A more extended description and comparisons including a better Figure 7 will benefit your paper. I suggest a close-up of the photographs in Fig. 7 (without the bones labelled in the photograph) and also including interpretative drawings for both cases with the main structures and bones labelled.

As mentioned above, there are several comments related to the language so I recommend you to check the text by a fluent English speaker before re-submission.

Thank you for submitting your manuscript to PeerJ and I look forward to receiving your revision.

Sincerely,


Claudia Marsicano

·

Basic reporting

This paper represents an important new finding of the rare dicynodont genus Jimusaria, at another Chinese locality but one geographically and stratigraphically far removed from what was previously known. The description is generally thorough and for the most part I agree with the anatomical conclusions. The language needs editing for grammar, for some examples see "Additional Comments" below, but the entire text should be carefully copy-edited for resubmission.

Experimental design

The research question, which is fundamentally about diversity and basinal distributions in time and space, is one that is currently quite vital to our understanding of turnover in the late Permian and around the Permo-Triassic mass extinction. The discovery of regional species-level diversity in China parallels that already demonstrated for African basins, indicating that similar forces driving speciation were operating globally at the time. The new specimens provide useful anatomical insights for the genus Jimusaria, which was previously known only from a single good and several poorly-preserved skulls.

Validity of the findings

I agree with the authors that the specimens described here are both referable to the genus Jimusaria and represent a new species. I am skeptical of a couple of the anatomical conclusions, specifically fusion of the nasals. I will admit I find the epipterygoid rod as shown in this paper to be very strange. It is very different from what I have seen in other dicynodontoids, to the degree that I almost wonder if it could be some sort of taphonomic distortion, with a portion of the epipterygoid preserved out-of-place. I presume the authors have already considered and excluded this possibility, but it might be worthwhile to address that explicitly in the text.

Additional comments

Line 1: The statement at the end of the title (“with the unique epipterygoid”) is awkward (especially as it refers to both species of Jimusaria, not just the new species). I would recommend deleting it.
Line 52: Daqingshanodon is misspelled.
Line 86: Recent study indicates that the authorship of Dicynodontia can actually be attributed to Owen, 1859 (Reference: Owen, R. 1859. On some Reptilian Remains from South Africa. The Edinburgh New Philosophical Journal. 10: 289-291.)
Line 94: “nasals fused as single element”—I am not sure how you would be certain of this given the nature of the available fossils. Many dicynodontoids have indistinct sutures between the nasals because of the nasal boss, furthermore the bone surface preservation of most of the Jimusaria sinkianensis specimens is insufficient to show whether fusion has occurred. I would consider this character questionable. CT-data could help demonstrate whether there are sutures extending throughout the internal contacts of the bones, but even then the material may not be well-preserved enough to recognize them.
Line 112: Change “has weathered roof” to “has a weathered roof”
Line 148: Change “of pineal foramen” to “of the pineal foramen”
Line 149: Change sentence to “Of the preserved postorbital bars, the only complete one is on the left side of IVPP V31929.”
Line 155: Change sentence to “The poor preservation obscures whether a distinct sagittal crest was present.”
Line 167: Change “of choana” to “of the choana”
Line 176: Change “of palatine pad” to “of the palatine pad”
Lines 178–179: This is confusing, do you mean that the ventral ridge on the anterior pterygoid ramus extends for most of the length of the ramus?
Line 185: Change “of basal tuber” to “of the basal tuber”
Line 217: Change “in exoccipital” to “surrounded by the exoccipital”
Line 225: Change sentence to “A distinct but thin lateral dentary shelf is situated on the dorsal edge of the mandibular fenestra, extending for almost the entire length of the fenestra and expanding into a prominent swelling anterodorsally.”
Line 241: Change “preserved” to “preserves”
Line 247: Change sentence to “The odontoid is partly exposed in dorsal view, fused with the axial centrum.”
Line 256: Change “In previous study” to “In a previous study”
Line 266: Change “from new species” to “from the new species”
Line 270: Delete “In” at the beginning of the sentence.
Line 317: Change “show similar structure in comparable portion” to “show a similar structure in a comparable portion of the skull”
Line 319: “epipterygoid” is misspelled
Line 330: Change “horizon orientation of stapes” to “horizontal orientation of the stapes”
Line 342: Change comma to semicolon
Line 375: Change “diversity” to “diverse”
Line 392: Change “dicynodonts” to “dicynodont”

Reviewer 2 ·

Basic reporting

This is a thorough and professionally written description of a new dicynodont species from the Naobaogou Formation. I appreciate the thoughtfulness with which the authors made comparisons with the other species of Jimusaria and how well they explained and illustrated the unique features of the new species. The figures are very well-done. The writing is concise and still gets the main points across.
I would suggest thinking about a short paragraph to insert at the beginning about the evolutionary/biogeographic significance of either Jimusaria, dicynodonts, or the fossil assemblages of Nei Mongol and/or Xinjiang. This would be good to introduce the importance of the subject to your readers and give context for this study.
I uploaded a pdf with some small edits throughout the paper. There are some connecting words missing in some sentences and a few sentences that are ambiguous that I pointed out. Also, there were a few times when the word “epipterygoid” was misspelled without the Y, so you should search through the manuscript for any remaining “epiptergoid”s and add the Y in. Other than these few spots, the paper is very well written and easy to understand.

Experimental design

This paper does a really thorough job of comparing the new species to Jimusaria sinkianensis, but I think the description also needs more comparisons to other dicynodonts and possibly other therapsids. A lot of the features described are probably shared with other species and may be phylogenetically or functionally significant, so it is good practice to make notes of those similarities throughout the description. Use more phrases like, “X bone has X morphology, as seen in X other dicynodonts (citations).” A specific example I noticed was in describing the ridge from the angular to the surangular. This is a common feature among dicynodonts, so you could point that out and maybe cite papers that talk about this feature in other species.
This is mostly out of curiosity but maybe something for you to think and talk about, but is there a reason other dicynodonts wouldn’t need the extra stability of the epipterygoid rod that Jimusaria has? Would other dicynodonts benefit from it or would it only be useful to an animal with the feeding morphology of Jimusaria? I just find it interesting that this feature is not seen in other dicynodonts, and other readers will likely wonder about this, as well.

Validity of the findings

I really like how the authors handled ontogeny in the discussion paper by comparing ontegenetic trajectories between the two species of Jimusaria. This really strengthens the argument for why this is a new species. In general, I think the conclusions of this paper are well-supported.
Line 329: What about the shape of the nasofrontal suture makes you think Jimusaria could not have been fossorial? I think you should include a short (1 sentence or partial sentence) explanation of why their anatomy doesn’t fit with a fossorial lifestyle. You should probably do this for the stapes as well, as people might not know what the significance of a horizontal stapes is. Maybe cite the papers by Laaß on the stapes orientation in other dicynodonts.
Line 230: Dicynodonts always have at least a slightly concave surface on the ventral half of the reflected lamina, called the "posteroventral fossa" by Olroyd and Sidor, 2022. Figure 5b seems to show this fossa, so you should double check this specimen and follow their terminology here.

Annotated reviews are not available for download in order to protect the identity of reviewers who chose to remain anonymous.

·

Basic reporting

In this manuscript the authors have done an excellent work on clearly and accurately describing the anatomy of a new species of dicynodont, Jimusaria monanensis. This is an important contribution as the new taxon adds to our knowledge of the Naobaogou Formation tetrapod diversity and strengthens the biostratigraphic correlation between the Nei Mongol and Xinjiang faunal associations. I am eager to see it published in the PeerJ journal.

The text is in general terms well-written, although it would benefit from the editing of a fluent English speaker. The work is properly contextualized, and, to my knowledge, references are sufficient and updated. The structure of the article seems to conform to the journal guidelines and all the relevant data to replicate the phylogenetic results is available as supplementary files. The figures are relevant and sufficient. However, I think that labeling some structures and adding a line drawing of the postcranial elements would be valuable. In particular, pointing to the contact between the epipterygoid and the braincase in figure 7A is crucial as this contact is discussed in depth along the manuscript.

Experimental design

This is an original research article worth of publication. The research question, mainly the description of a new species in a phylogenetic context, is well-defined, relevant and meaningful for the scientific community. It is clearly stated how this contribution fills an identified knowledge gap. The authors present detailed descriptions and comparisons. The methods are described with enough detail for the phylogenetic analysis to be replicated.

Validity of the findings

The results are important for the scientific community.
Conclusions are pertient.

Additional comments

In my opinion, this manuscript is very interesting and worth of publication. Minor corrections, suggestions and comments are to be found in the reviewed manuscript file.

---

## Round 0.2 · accepted · Accept

Dear Dr. Shi

I am pleased to inform you that your manuscript # 86077 entitled "The tetrapod fauna of the upper Permian Naobaogou Formation of China: 10. Jimusaria monanensis sp. nov. (Dicynodontia) with the unique epipterygoid", co-authored with Jun Liu, is now accepted for publication in PeerJ.

Thank you again for considering PeerJ and we look forward to your future contributions to the Journal.

sincerely,

Claudia Marsicano